# Upgrading the Functional Potential of Apple Pomace in Value-Added Ingredients with Probiotics

**DOI:** 10.3390/antiox11102028

**Published:** 2022-10-14

**Authors:** Camelia Cristina Vlad, Bogdan Păcularu-Burada, Aida Mihaela Vasile, Ștefania Adelina Milea, Gabriela-Elena Bahrim, Gabriela Râpeanu, Nicoleta Stănciuc

**Affiliations:** Faculty of Food Science and Engineering, Dunărea de Jos University of Galati, Domnească Street 111, 800201 Galati, Romania

**Keywords:** apple pomace, probiotics, prebiotics, pectin, value-added, polyphenols, circular economy

## Abstract

Emerging customized designs to upgrade the functional potential of freeze-dried apple pomace was used in this study, in order to transform the industrial by-products into ingredients containing probiotics, for a better and healthier food composition. The freeze-dried apple pomace was analyzed for free and bounded phenolic contents, highlighting a significant level of caffeic acid (4978.00 ± 900.00 mg/100 g dry matter (DM)), trans-cinnamic acid (2144.20 ± 37.60 mg/100 g DM) and quercetin 3-β-D-glucoside (236.60 ± 3.12 mg/100 g DM). The pectin extraction yield was approximatively 24%, with a degree of esterification of 37.68 ± 1.74%, and a methoxyl content of 5.58 ± 0.88%. The freeze-dried apple pomace was added in a different ratio as a supplement to cultural medium of *Loigolactobacillus bifermentans* MIUG BL 16, suggesting a significant prebiotic effect (*p* < 0.05) at concentration between 1% and 2%. The apple pomace was used to design three freeze-dried ingredients containing probiotic, with a high level of polyphenolic content (6.38 ± 0.14 mg gallic acid equivalents/g DM) and antioxidant activity (42.25 ± 4.58 mMol Trolox/g DM) for the powder containing apple pomace ethanolic extract. When inulin was used as a prebiotic adjuvant, the obtained powder showed a 6 log/g DM viable cell count. The ingredients were added to fermented vegetable soy milk-based products, allowing us to improve the polyphenolic content, antioxidant activity and viable cell counts. The approach designed in this study allowed us to obtain ingredients suitable to add value to food, whereas premises to align with the current circular economy premises, by reintegrating the industrial waste as sources of high added value compounds, are also provided.

## 1. Introduction

The actual projection indicates a continuous population growth, which will probably reach the level of about 10 billion in 2050; this imposes a high pressure on the agri-food market to offer alternative sources of food, which should meet the nutritional needs and also to the current trend of healthy and sustainable foods consumption [1,2]. However, huge quantities of food wastes are generated by the agro-food industries, resulting in a negative impact on environment, due to their high moisture content and instability, thus favoring its microbial decomposition with a concomitant production of greenhouse gas emissions and nitrogen contamination of soil and water [3]. An estimation of agro-food waste during production and processing of fruits and vegetable reaches approximately 14.8%, accounting for the largest source of food loss and wastes globally [4], involving significant economic and environmental issues. Commonly, these agri-food wastes are used as animal feed, landfilling, composting, anaerobic digestion, carbonization or thermal treatment [5]. Recent strategies for reducing or recycling agri-food waste have reconsidered integrated approaches for the production of bioenergy, biochemicals, and value-added products, driven by diversity and high concentration of compounds and secondary metabolites, with multiple biological and technological functions [6]. Fruit and vegetable wastes are rich in non-digestible carbohydrates (resistant starch, inulin, cellulose, hemicellulose, pectin and alginates), polyphenols, carotenoids, vitamins, etc. [6].

The apple (*Malus domestica*) is one of the most consumed fruits worldwide, both as fresh and processed products. In order to meet the global demands for juices, juice concentrates and cider, 11.6 million tons of apples are processed [7], resulting in 30% of the product becoming wastes. Globally, these wastes may represent up to 4 million tons per year of apple pomace [6], consisting of pulp, skins, seeds and stalks of the fruit [8]. The apple pomace is rich in minerals, dietary fiber, polyphenols, and pectin [9,10]. Two important recycling strategies for fruit and vegetable wastes were reported, divided into different technologies, such as composting, processing to flour or conversion into water, and extractions [11]. Therefore, the application of apple pomace could be divided into two main ways, respectively, conventional (such as an additive in animal feed, fermentation conversion to compost, or produce nutrition enhancement) and high-value products (functional ingredients exploring the carbohydrates, phenolic compounds and pentacyclic triterpenes content, and extraction or fermentation conversion to produce enzymes, organic acid, pigment, biofuels) [12]. The application of selected extraction to obtain high-value-added ingredients may be considered as an economically and environmental efficient strategy, since novel extraction technologies of a low-cost, easily available material guarantee a high extraction rate and yield, while reducing the use of organic solvents [11]. However, when designing a technology to extract selected bioactives based on affordable, sustainable and profitable technologies at an industrial level [13], several cost-related issues should be considered such as the costs of initial investment, the industrial application, the disposal of the relatively high amounts of residual waste, etc. [13].

The strain-specific probiotic bioactivities are extensively studied, especially in vitro, whereas very few have tested probiotic efficacy in an animal or human model [5]. The health-associated benefits of probiotics are related to increasing the food’s nutritional value, intestinal infections regulation, lactase biosynthesis to improve lactose digestion, anti-cancer property and antibiotic therapy, and reducing diarrhea incidence and blood cholesterol [14,15]. The survivability of probiotics is significantly questioned, since the extrinsic and native features such as the production of hydrogen peroxide, post-acidification, oxygen, pH, storage temperature, and processing conditions can greatly reduce their activity [16,17]. In order to improve the probiotics viability in different environments, different prebiotics, such as oligosaccharides, mannan oligosaccharide, galactooligosaccharide, arabinoxylan-oligosaccharide, inulin, and β-glucan, have been well studied [5]. The prebiotic mechanism is explained by their slow fermentation from complex structures, thus providing fermentable carbohydrates for bacteria in the distal colon, allowing us to regulate the dysbiosis of gut microbiota, while producing metabolic butyrate to modulate the gut barrier function and anti-inflammatory effect [18]. The prebiotics targeted for the gut should be resistant to gastrointestinal digestion, including the low pH of the stomach, hydrolysis by intestinal enzymes and gastrointestinal absorption [19]. It has been suggested that approximately 85–90% of the ingested pectin reached the terminal ileum [20], being available for microbial fermentation at the colon.

The vital role of the lactic acid bacteria (LAB) in human health [21] is well studied, with a significant dimension of industrial applications, both in the health and agri-food industries to enhance food quality and human well-being [22]. Recently, the LAB taxonomy included 261 species, divided into 26 genera based on their whole genome sequences [23]. Significant scientific data support the benefits of lactic acid bacteria as probiotics, modulating the gut microbiota due to their ability to compete with pathogens, as well as their immunomodulatory, anti-obesity, anti-diabetic, and anti-cancer activities [14]. *Loigolactobacillus bifermentans* (*Lo. bifermentans*) is a facultatively heterofermentative lactic acid bacterium, generally isolated from cheeses, with a potential for fermentation of lactic acid into acetic acid, ethanol, traces of propionic acid, carbon dioxide, and hydrogen [24].

Therefore, the aim of our study was to test the hypothesis that apple pomace is suitable as a potential valuable resource for full-components utilization, in terms of polyphenols and prebiotic fibers, in order to design novel formulations based on apple pomace and probiotics [6]. In this context, the pomace resulted from apple juice (Idared variety) was selected as an emerging source of polyphenols, and pectin as new prebiotics for the selected LAB strain. The apples pomace was freeze-dried and used to advance the phytochemical content, in terms of polyphenols and pectin. Further, in order to customize the technological design, the freeze-dried apple pomace was tested for prebiotic potential for *Loigolactobacillus bifermentans* MIUG BL 16 (*Lo. bifermentans* MIUG BL 16). The strain was isolated from cheese and conserved with the indicative of MIUG 16 in the Microorganisms Collection of University Dunarea de Jos of Galati, Romania. Three customized ingredients were developed based on apple pomace and *Lo. bifermentans* MIUG BL 16, with different adjuvants (inulin and soy protein isolates), in order to maintain unique attributes such as shelf-life and probiotic cell viability to fulfill the specific needs of dosage. The resulting powders were tested for phytochemical content, in terms of polyphenols and flavonoids content, antioxidant activity, and viable cell counts, whereas the value-added functional properties were analyzed by adding to vegetable fermented foods.

## 2. Materials and Methods

### 2.1. Chemicals

The HPLC analytical-grade hexane, acetone, acetonitrile, ethyl acetate, methanol and analytical grade 2,2-diphenyl-1-picrylhydrazyl (DPPH), 6-Hydroxy-2,5,7,8-tetramethylchromane-2-carboxylic acid (Trolox), hydrochloric acid, citric acid, sodium hydroxide, sodium chloride, aluminum chloride, ethanol, Folin–Ciocâlteu reagent, gallic acid, inulin were purchased by Sigma Aldrich (Darmstadt, Germany). For cromatographic analysis the following reagents were used: HCl ACS reagent (37%), acetic acid, methanol, ethyl acetate, acetonitrile, theaflavin, cafestol, procyanidin A1, procyanidin B1, (−)-epigallocatechin, catechin, caffeine, caffeic acid, ellagic acid, gallic acid, protocatechuic acid, trans-cinnamic acid, quercetin 3-glucoside, quercetin 3-D-galactoside, quercetin 3-β-D-glucoside, naringin, hesperidin, myricetin, apigenin, kaempferol, luteolin, and isorhamnetin (HPLC-grade), purchased from Sigma-Aldrich (Darmstadt, Germany). Other reagents such as sodium bicarbonate were purchased from Honeywell, Fluka (Seelze, Germany). The *Lo. bifermentans* MIUG BL 16 strain was from Microorganism Collection of Dunarea de Jos University (acronym MIUG, Galati, Romania). de Man, Rogosa and Sharpe agar (MRS agar) was purchased from Merck (Darmstadt, Germany).

### 2.2. Fruits Processing

Apples from Idared variety were purchased from a local market (Galati, Romania) in October 2021 and washed. The seeds and stems were removed manually, and the apples were portioned into small slices (approximatively 5 cm). Further, in order to limit the oxidative processes, the slices were immersed in a solution consisting of lemon juice and honey (1000 mL of distillated water, 10 mL of lemon juice and 10 g of honey) for 4 h. The composition of the apple’s immersion solution took into account the use of natural sources of ascorbic acid and sugars, such as lemon juice and honey. Ascorbic acid is a common antioxidant that can rapidly reduce quinones and inhibit enzymatic browning [25]. The apple slices were squeezed, using a fruits juicer (Stainless Steel Fruit Vegetable Juice Extractor Juicer Squeezer, Guangdong, China). The resulting pomace was immediately frozen (−18 °C) and subjected to freeze-drying (CHRIST Alpha 1-4 LD plus, Osterode am Harz, Germany) at −42 °C under a pressure of 0.10 mBar for 48 h. Afterwards, the freeze-dried pomace was collected and packed in metallized bags and stored at 4 °C until further analysis.

### 2.3. Conventional Solid–Liquid Solvent Ultrasound Assisted Extraction of Polyphenols from Freeze-Dried Apple Pomace

In order to have an overall view of the global polyphenols content, a conventional solid–liquid solvent ultrasound assisted method was applied. Therefore, an amount of 1 g of freeze-dried apple pomace was homogenized with 25 mL of 70% ethanol solution and extracted in an ultrasonic bath, at 35 °C, for 30 min. Afterwards, the mixture was centrifuged at 3420× *g* for 10 min, at 10 °C (Hettich Universal 320R, Tuttlingen, Germany), the supernatant was collected, and the extraction was repeated three times. The collected supernatants were subjected to concentration, at the temperature of 40 °C, under reduced pressure to dryness (AVC 2-18 concentrator, CHRIST, Osterode am Harz, Germany). The obtained extract was used for the spectrophotometric analysis of total polyphenolic (TPC) and flavonoids contents (TFC).

### 2.4. Spectrophotometric Analysis of Total Polyphenolic and Flavonoids Contents from Freeze-Dried Apple Pomace

To measure the content of TPC and TFC, the colorimetric method were used. For TFC content analysis, the aluminum chloride method was used, involving the mixing of 0.25 mL of a solution consisting of 10 mg of concentrated extract dissolved in 5 mL of ultrapure water, with 1.25 mL of distilled water and 0.075 mL of 5% sodium nitrite solution. After 5 min of reaction, at room temperature, a volume of 0.150 mL of 10% aluminum chloride solution was added and allowed to react for 6 min [26]. Further, a volume of 0.5 mL of 1M sodium hydroxide and 0.750 mL of distilled water were added and the absorbance of the mixtures was immediately measured at 510 nm (Jenway 6505 UV-Vis spectrophotometer, Loughborough, UK). The TFC content was expressed in mg catechin equivalents/g dry matter (mg CE/g DM), based on catechin standard curve.

The Folin–Ciocâlteu method was used to determine the TPC content, by using a volume of 0.2 mL extract dissolved in ultrapure water, mixed with 15.8 mL of distilled water and 1 mL of Folin–Ciocâlteu reagent [26]. The mixture was allowed to react for 10 min, at room temperature, followed by addition of 3 mL of 20% sodium carbonate solution. The absorbance was measured after 60 min of reaction in the dark at a wavelength of 765 nm (Jenway 6505 UV-Vis spectrophotometer, Loughborough, UK). The TPC content was expressed as mg gallic acid equivalents (GAE)/g DM, based on gallic acid standard curve.

### 2.5. Extraction of Free and Bounded Polyphenols from Freeze-Dried Apple Pomace

The free and bounded bioactive compounds’ extraction and separation was performed as described by Xiong et al. [27], slightly modified as Zhang et al. [28]. Briefly, the freeze-dried apple pomace’s free bioactives were extracted with methanol (80% *v*/*v*), at 25 °C and 150 rpm, for 2 h (Lab Companion SI-300, GMI, Washington, DC, USA), the supernatant being concentrated afterwards. For the bounded bioactive compounds from the analyzed sample, the pellet resulted after free bioactives’ extraction was mixed with 2M HCl and hydrolyzed at 100 °C for 1 h. Thereafter, the bounded bioactive compounds were extracted with ethyl acetate and then concentrated. The obtained fractions were used for chromatographic analysis of the polyphenols.

### 2.6. Cromatographic Analysis of Polyphenols from Freeze-Dried Apple Pomace

The separation and identification of the bioactive compounds from freeze-dried apple pomace was performed by an Agilent 1200 HPLC system equipped with autosampler, degasser, quaternary pump system, multi-wavelength detector (MWD) and column thermostat (Agilent Technologies, Santa Clara, CA, USA). A Synergi Max-RP-80 Å column (250 × 4.6 mm, 4 µm particle size, Phenomenex, Torrance, CA, USA) was used for the free and bounded polyphenols analysis. The concentrated samples, free and bounded fraction, respectively, were dissolved in methanol and subjected to the HPLC separation, at 30 °C, using 10 µL injection volume, with solvent A (1% *v/v* acetic acid in ultrapure water) and solvent B (acetonitrile) using a flow rate of 0.65 mL/min. The method runtime was 80 min, the compounds of interest were detected at 280 nm and 320 nm, using a 10 nm bandwidth for each detection wavelength to increase compounds’ selective separation. The identification of the bioactive compounds from apple pomace was made by comparing the retention times of the peaks with those obtained for standard solutions of bioactives. The identified compounds were quantified by external calibration curves using the peak area. Data acquisition was performed by Chemstation software, version B.04.03 (Agilent Technologies, Santa Clara, CA, USA). Results were expressed in mg/100 g DW freeze-dried pomace.

### 2.7. Pectin Analysis

The citric acid method proposed by Spinei and Oroian [29] was used for the extraction of pectin from freeze-dried apple pomace. Briefly, the freeze-dried apple pomace was mixed with ultrapure water (1:10 *w*/*v*), the pH of this mixture being adjusted to 2.0, and then kept, at 90 °C, for 3 h in a water bath (Julabo, Seelbach, Germany). Thereafter, the supernatant was separated by centrifugation (2320× *g*, 22 °C, 35 min) followed by an equal amount of ethanol (96% *v*/*v*) addition (1:1 *v*/*v*). The precipitated pectin was separated after 12 h of storage, at 4 °C, and centrifugation in the same conditions. Finally, the extracted pectin was washed with ethanol (96% *v*/*v*) and dried at 50 °C. The extraction yield was expressed using Equation (1):(1)$$Pectin\;yield,\;\%=\frac{m_{0}}{m}\times 100\;$$
where *m*_0_ is the weight of dried pectin (g), and *m* is the weight of dried pomace (g).

The degree of esterification for the extracted pectin was assessed by the titrimetric method using NaOH 0.10 N and phenolphthalein [30]. Results were expressed as described in Equation (2):(2)$$Degree\;of\;esterification,\;\%=\frac{V_{2}}{V_{1}+V_{2}}\times 100\;$$
where *V*_1_ is the volume of NaOH 0.10 N used for the first titration (mL), and *V*_2_ is the volume of NaOH 0.10 N used for the second titration (mL).

Furthermore, the equivalent weight (EW) for the pectin extracted from the freeze-dried apple pomace was determined. Firstly, for the EW estimation, 0.25 g pectin was dissolved in 100 mL ultrapure water and continuously mixed (300 rpm, 1 h, 25 °C). The EW was evaluated by the titrimetric method with NaOH 0.10 N, and prior to the titration, 5 g of NaCl and phenolphthalein were added in the pectin solution. Results were calculated using Equation (3) [30]. This resulting solution was further used to determine the methoxyl content by titration with NaOH 0.10 N as Dranca et al. [31] reported. Methoxyl content was calculated by Equation (4).
(3)$$EW,\;g/mol=\frac{1000\times m}{V\times N}\;$$
where *m* is the sample weight (g), *V* is the volume of NaOH 0.10 N used for titration (mL), and *N* is the normality of NaOH solution.
(4)$$Methoxyl\;content,\;\%=\frac{V\times N\times 3.10}{m}$$
where *V* is the volume of NaOH 0.10 N used for titration (mL), *N* is the normality of NaOH solution, and *m* is the sample weight (g).

### 2.8. Antiradical Scavenging Activity

The antiradical activity of the extract on 2,2-diphenyl-1-picrylhydrazyl (DPPH) was used to evaluate the antioxidant activity of the apple pomace. The protocol involved mixing a volume of 3.9 mL of DPPH solution (0.1 M in methanol) with 0.1 mL of extract solution dissolved in ultrapure water. The mixture was allowed to react for 90 min, at room temperature, in the dark, followed by absorbance reading at 515 nm (Jenway 6505 UV-Vis spectrophotometer). The antiradical activity was expressed as mMol Trolox/g DM, based on 6-Hydroxy-2,5,7,8-tetramethylchromane-2-carboxylic acid (Trolox) standard curve [26].

### 2.9. Lo. bifermentans MIUG BL 16 Reactivation and Inoculum Preparation

Prior to each testing and formulation, all the growth media were autoclaved, at 121 °C, for 15 min (ES-315, TOMY Digital Biology Co., Ltd., Tokyo, Japan), whereas the freeze-dried apple pomace was sterilized using a UV lamp (Faster SafeFast Elite, Cornaredo, Italy). The bacterial *Lo. Bifermentans* MIUG BL 16 strain stored in MRS broth medium with 40% glycerol, at −80 °C, in MIUG Collection was reactivated in MRS broth medium overnight and incubated for 48 h, at 37 °C, followed by being spread plated in the polystyrene Petri dishes plates containing the MRS agar surface. The inoculum was obtained from a single colony harvested from MRS agar, transferred to 50 mL MRS broth medium, and cultured for 48 h, at 37 °C. The optical density of the inoculum was 2.0 at a wavelength of 600 nm, using a Jenway 6505 UV-Vis spectrophotometer.

### 2.10. The Prebiotic Effect of Freeze-Dried Apple Pomace

In order to test the prebiotic effect, different ratios of the freeze-dried apple pomace (0.5%, 1.0% and 2.0%, *w*/*v*) were added to 98 mL of liquid MRS medium, followed by addition of 2% (*v*/*v*) inoculum. The control sample obtained without the addition of apple pomace was prepared using the same conditions. The samples were fermented in a stationary system at, 37 °C, for 48 h. After fermentation, the samples were stored, at 4 °C, for 14 days. The prebiotic effect was calculated based on Equation (5):(5)$$Prebiotic\;effect=\frac{log{\left(\frac{CFU}{mL}\right)}_{sample}}{log{\left(\frac{CFU}{mL}\right)}_{control}}\;$$
where *log (CFU)/mL_sample_* represents the logarithm of the colony-forming units of the sample with the addition of freeze-dried apple pomace, and *log (CFU)/mL_control_* represents the logarithm of the colony-forming units of the control sample (without the addition of apple pomace).

### 2.11. Viable Cell Counts

In order to determine the viable cell counts, the method described by Vasile et al. [32] was used by 10-fold serial dilutions in a sterile physiological serum (0.9 g NaCl%, *w*/*v*), by the pour plate technique. The viable cell number was determined by estimating the number of colony-forming units (CFU) on the MRS agar plates (medium at pH 5.7) after 72 h of aerobic incubation, at 37 °C. The counts were expressed as CFU/g DM.

### 2.12. Customised Carriers for Inoculation of Freeze-Dried Apple Pomace with Lo. bifermentans MIUG BL 16

Three experimental variants were prepared based on freeze dried apple pomace and 1% *L. bifermentans* MIUG BL 16 inoculum, as follows: Variant 1 (V1) was obtained by using a single combination of freeze-dried apple pomace dissolved in sterile distillated water (20%) and probiotic culture, Variant 2 (V2) combined freeze-dried apple pomace dissolved in sterile distillated water (20%) with 2% inulin and probiotic culture, whereas Variant 3 (V3) used an amount of 5 g of the concentrated extract obtained from freeze-dried apple pomace, dissolved in biopolymeric solution of soy protein isolate (2%) and inulin (1%) and probiotic culture. All the samples were immediately frozen (−18 °C) and subjected to freeze-drying (CHRIST Alpha 1-4 LD plus, Osterode am Harz, Germany), at −42 °C, under a pressure of 0.10 mBar for 48 h. Afterwards, the carriers were collected and packed in metallized bags and stored, at 4 °C, until further analysis.

### 2.13. Global Characterization of Powders

The powders were extracted for phytochemicals, and characterized as TPC, TFC, antioxidant activity, and viable cell counts as described according to the protocols abovementioned.

### 2.14. Testing the Powders in Food Matrices

The powders were tested for value-added potential by adding into a fermented soya milk product in a ratio of 2%. The samples were coded as C (control), S1 (fermented soya milk products with 2% addition of V1), S1 (fermented soya milk products with 2% addition of V2), and S3 (fermented soya milk products with 2% addition of V3). After addition, the resulting samples were kept for 24 h under refrigeration conditions, for uniformity and homogenization. The samples were characterized for TPC, TFC, antioxidant activity and viable cell counts. In order to assess the cells viability during storage, the foods were analyzed after 7 and 14 days of storage in refrigerated conditions.

### 2.15. Statistical Analysis

Reported results are average values for triplicates (*n* = 3) and duplicate measurements (*n* = 2) in case of HPLC data, followed by standard deviations. Significant differences were determined by ANOVA and Tukey test for 95% confidence interval (*p* < 0.05) using Minitab 19 software (Minitab Inc., State College, PA, USA).

## 3. Results and Discussion

### 3.1. Phytochemical Content of the Freeze-Dried Apple Pomace

Identification of phenolics in the extracts obtained from freeze-dried apple pomace was achieved by comparing their spectra and retention times with those of externally injected standards. Several compounds for which standards were not available, and identification was made using references. Figure 1 shows the detected free phenolics analyzed in freeze-dried apple pomace, at different wavelength (280 nm and 320 nm). A total of 52 and 44 phenolic substances were separated through retention time (Rt), respectively (Figure 1a,b).

Six classes of free compounds were identified and quantified, based on external calibration curves (Table 1). The total free polyphenolic content, estimated based on quantity of identified compounds were 134.96 mg/100 g DM pomace at 280 nm and 137.52 mg/100 g DM pomace. When measuring at 280 nm, the flavan-3-ols were identified as the major polyphenolic class (79.12%), followed by flavanones (10%) and phenolic acids (6.93%). When measuring at 280 nm, fourteen free polyphenols were identified and quantified, according to the reference standards, as follows: caffeic acid, ellagic acid, protocatechuic acid, theaflavin, catechin, cafestol, procyanidins A1 and B1, (−)-epigallocatechin, caffeine, quercetin 3-glucoside, hesperidin and naringin. However, at 320 nm, gallic acid, caffeic acid, myricetin, quercetin 3-glucoside and hesperidin were found (Table 1). The major free bioactive were represented by gallic acid (86.42 ± 2.26 mg/100 g DM), theaflavin (71.73 ± 0.72 mg/100 g DM) and hesperidin (42.83 ± 0.27 mg/100 g DM).

In case of bounded phenolics, from Figure 2a,b, it can be observed that 44 and 24 phenolic substances were separated through retention time (Rt), at 280 nm and 320 nm, respectively. The total concentration for the bounded phenolics measured at 280 nm was of 5362.71 mg/100 g DM apple pomace, whereas at 320 nm, a concentration of 2299.21 mg/100 g DM apple pomace was found. From Table 1, it can be seen that caffeic acid prevails, with the highest concentration of approximatively 50 mg/g DM (49.78 ± 9.00 mg/g DM), followed by trans-cinnamic acid, with 21.44 ± 0.37 mg/g DM and quercetin 3-β-D-glucoside, with 2.36 ± 0.03 mg/g DM.

The phenolic acids were the main bounded compounds, representing 94% from the total polyphenolics. Different studies reported that in fresh apple pomace, phenolic compounds are dominated by chlorogenic acid, caffeic acid, (+)-catechin, (−)-epicatechin, rutin, and quercetin glycosides [33], whereas after freeze-drying, phlorizin (phloretin-2′-β-D-glucopyranoside) is the most prominent polyphenol [34].

Additionally, Górnaś et al. [35] suggested that di-hydrochalcones (such as phlorizin) are the main polyphenols found in apple seeds and stems, whereas the flesh part consists mainly of chlorogenic acid and flavonol glycosides. However, in this study, the seeds and stems were removed in the apples processing steps, and therefore, it is not expected to identify and quantify these compounds.

### 3.2. Pectin Quantification and Characterization from the Freeze-Dried Apple Pomace

The citric acid method used in our study allowed us to obtain a pectin yield of 23.62 ± 0.70%, with a degree of esterification of 37.68 ± 1.74%, EW of 149.26 ± 1.89, and a methoxyl content of 5.58 ± 0.88%. Our results are in good agreement with Colodel & Petkowicz [36], who reported a pectin yield of 27.40% after 1 h of extraction with citric acid. It has been suggested that the pectin yield may be both positively and negatively affected by the preliminary treatments applied to the pomace, as well as by the pH and type of solvent, extraction time and temperature [37]. Pectin yields ranging 21.49–22.17% were determined after apple pomace’s treatment, at 80 °C, for 1 or 2 h [38].

Zheng et al. [39] reported a degree of esterification of 34.60% for the pectin extracted from apple pomace, which is in fair agreement with data reported in this study (37.68 ± 1.74%). However, it has been suggested that a degree of esterification higher than 50% can be achieved by a statistical optimization of the pectin’s extraction process. The methoxyl content (5.58 ± 0.88%) and EW values (149.26 g/mol) can be correlated with extraction parameters, such as pH and temperature, whereas some other parameters (such as citric acid and extraction time) affect the monosaccharides’ structures of pectin, leading to a low value for EW [31]. Spinei and Oroian [29] reported methoxyl content ranging from 3.98 to 6.18% for the pectin extracted from grape pomace.

### 3.3. Potential Prebiotic Effect of Freeze-Dried Apple Pomace Supplementation of Broth Culture Medium

Different studies showed that the chemical compounds of apple pomace, in terms of pectin and polyphenols, make this by-product suitable for valorization as a potential prebiotic [5]. For example, different studies suggested that apple pectin is able to interact both with the intestinal microbiota and the intestinal immune cells, thus acting as a prebiotic substrate capable of promoting the intestinal immune barrier [40]. In this study, the MRS broth was supplemented with different ratios of freeze-dried apple pomace (0–2%, *w*/*v*), followed by inoculation with *Lo. bifermentans* MIUG BL 16 (2%) and fermentation under controlled stationary system (37 °C for 48 h). In order to estimate the potential prebiotic effect, the viable cell counts were also determined after a 21 days storage, at 4 °C. The prebiotic effect of the supplementation is provided in Table 2.

From Table 2 it can be observed that after 7 days of storage, no significant differences (*p* > 0.05) were found in potential prebiotic effect of apple pomace supplementation of culture medium, whereas an increase was observed after 14 and 21 days, respectively. The potential prebiotic effect is much more obvious after 21 days of storage, at 4 °C, when a significant increase (*p* < 0.05) was observed at added concentration of 1 and 2% (Table 3).

The potential prebiotic effect of freeze-dried apple pomace could be explained by a synergic effect of both polysaccharides and polyphenolics. The prebiotic effect of polysaccharides is related to probiotics ability to degrade pectins, and/or utilize them and other metabolites, by cross-feeding interactions [41]. Larsen et al. [41] suggested that one of the most important factors in prebiotic effect of pectin is the degree of esterification of polygalacturonic acid, with an increase in microbiota composition for the low methoxyl pectins. Therefore, the significant prebiotic effect of freeze-dried apple pomace could be explained by the low methoxyl content for the pectin determined in this study (5.58 ± 0.88%). However, these authors highlighted the possibility of differential stimulation of bacterial populations using pectins with different sugar content [41].

Besides the well-recognized resistant oligosaccharides prebiotics (inulin, fructo-oligosaccharides and galacto-oligosaccharides) [42], recent studies have shown the interaction between polyphenols and the gut microbiota, suggesting them as candidate compounds to prebiotics [43,44]. For example, de Araùjo Chagas Vergara et al. [45] suggested the prebiotic effect of the fermented cashew apple juice containing oligosaccharides on *Leuconostoc mesenteroides* and *Lactobacillus johnsonii* growth.

Regarding the prebiotic potential of polyphenols, a bidirectional interaction was suggested, involving a modulation effect of polyphenols on gut microbiota and, conversely, microorganisms can modulate the activity of the phenolic compounds [46]. This bidirectional relationship can regulate the metabolism and the bioavailability of polyphenols, converting them into metabolites, which may have different effects on the host health, as explained by Singh et al. [44].

Although the prebiotic effect of apple pomace was clearly evidenced in this study, the potential role of polyphenols, sugars and pectin as modulators of gut microbiota should be further tested [47]. This is based on the hypothesis that the structure and function of each individual polyphenol can be influenced by the food matrix and individual composition of the human microbiota [48].

### 3.4. Customised Designs for Ingredients Based on Apple Pomace and Lactic Acid Bacteria

Three experimental variants of freeze-dried powders based on freeze-dried apple pomace and the corresponding extract and *Lo. bifermentans* MIUG BL 16 were obtained, with and without adjuvants. The powders were tested for global phytochemical content and viable cells (Table 3).

As expected, variant V3 containing apple pomace extract showed a higher TPC content of 6.38 ± 0.14 mg GAE/g DM powder, whereas the samples with apple pomace displayed comparable TPC values of 5.32 ± 0.06 mg GAE/g DM powder (V1) and 5.64 ± 0.09 mg GAE/g DM powder (V2). The same trend was observed in TFC, with lower value in V1 (4.02 ± 0.09 mg CE/g DM powder), followed by V21 with 5.21 ± 0.35 mg CE/g DM powder and V3 with 5.59 ± 0.22 mg CE/g DM powder. The higher TPC content in V3 had a significant impact on DPPH antiradical scavenging activity, yielding a value of 42.25 ± 4.58 mMol Trolox/g DM powder. The significant difference between variants could be explained by the use of the extract from apple pomace in designing V3, which concentrates various bioactive compounds, thus impacting the antioxidant activity. However, the customized design significantly impacted the viable cell counts (Table 3), with V2 and V3 reaching 6.30 log and 6.04 log, respectively, when compared with 5.40 log in V1. These results highlighted the cumulative effect of pectin, inulin and polyphenols in V2.

### 3.5. Testing the Value-Added Potential of the Ingredients in Food Matrice

In order to test the potential value-added functionality of the customized variants, a fermented commercial product obtained from soy milk was selected, from the perspective to improve the nutritional and biological profile of plant-based food products. The variants were added in a ratio of 2% and characterized for phytochemical content, antioxidant activity and viable cell counts (Table 4).

An increase in TPC and TFC contents was observed for all the sample, as compared with control (C), according to the profile of the powders. The value-added functional properties were highlighted in terms of both antioxidant activity and viable cell counts. The antioxidant activity values were significantly higher (*p* < 0.05) in variants S2 and S3 (37.95 ± 1.31 mMol Trolox/100 g DM product and 40.23 ± 1.27 mMol Trolox/100 g DM product, respectively), while viable bacterial cells showed higher values with 1 log, when compared with control (Table 4). Wang et al. [15] suggested polyphenol contents of 56.2 ± 1.8 µg GAE/g, 66.4 ± 3.4 µg GAE/g and 74.9 ± 2.6 µg EAG/g in yogurt samples fortified with 1%, 2% and 3% freeze-dried apple pomace, respectively.

## 4. Conclusions

The research carried out in this study is an attempt to answer the current demands to add value in food by the incorporation of natural ingredients. As apple is one of the most processed fruits worldwide, the immense amount of apple pomace generated is considered an important and cheap source of bioactives, with important benefits for health. Therefore, in our study, different customized designs to valorize apple pomace into ingredients were studied, as a valuable strategy to expand the food chain with less impact on the environment, while providing nutritional and biological added value in terms of bioactives, prebiotics and probiotics in single formulas. The apple pomace was used in a freeze-dried form to analyze the polyphenolic and pectin content. The preliminary chromatographic analysis evidenced the presence of both free and bonded polyphenolic compounds, allowing us to quantify a total amount of around 135 mg/100 g DM and 5363 mg/100 g DM freeze-dried apple pomace, respectively. Gallic acid, theaflavin and hesperidin were the major free compounds separated from the methanolic fraction, whereas caffeic acid showed the highest concentration of approximatively 50 mg/g DM, followed by trans-cinnamic acid, and quercetin 3-β-D-glucoside, these compounds being identified from the bounded fraction. However, an in-depth bioactives’ characterization along with their interactions and mechanisms of action must be considered to emphasize and understand the functionality of the designed products with apple pomace. The pectin yield was of approximatively 24%, with a degree of esterification of 38%. The prebiotic effect was tested for a probiotic strain (*Loigolactobacillus bifermentans* MIUG BL 16) in different ratios, varying from 0 to 2%, suggesting a cumulative effect of both pectin’s and polyphenols on cell survivability. Three customized technological designs were developed, incorporating freeze-dried apple pomace and inulin, whereas the extract was microencapsulated in a biopolymeric matrix based on soy protein isolates and inulin. The ingredients were analyzed for the phytochemical content and viable cell count of at least 6 log/g. The powders were added as ingredients in a fermented soy milk, allowing us to obtain an increase in antioxidant activity and viable cell counts. The results obtained in this study unveil a readily available method for straightforward processes in order to upgrade the great potential of fruit by-products by transforming them into value-added ingredients for different applications.

## Figures and Tables

**Figure 1 antioxidants-11-02028-f001:**
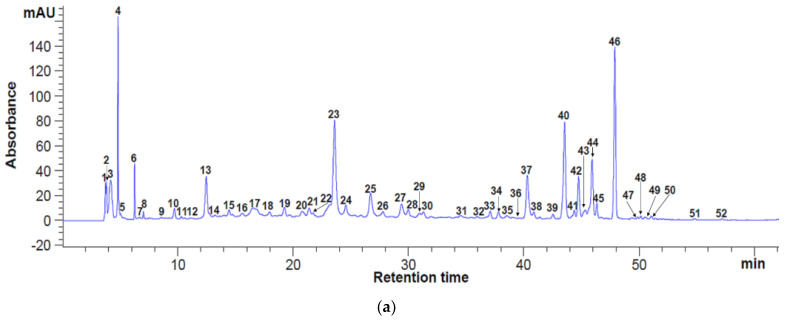
HPLC analysis for free polyphenolic compounds in freeze-dried apple pomace extract at 280 nm (**a**) and 320 nm (**b**). Peaks’ identification: (**a**): 4—theaflavin, 5—cafestol, 16—protocatechuic acid, 17—procyanidin B1, 19—(−)-epigallocatechin, 22—catechin, 24—caffeine, 26—caffeic acid, 31—procyanidin A1, 36—ellagic acid, 38—quercetin 3-glucoside, 41—naringin, 43—hesperidin, 1–3, 6–15, 18, 20, 21, 23, 25, 27–30, 32–35, 37, 39, 40, 42, 44–52—unidentified compounds; (**b**): 2–theaflavin, 3—cafestol, 5—gallic acid, 9—procyanidin B1, 14—caffeine, 16—caffeic acid, 19—procyanidin A1, 23—ellagic acid, 24—quercetin 3—D-galactoside, 25—quercetin 3—β-D-glucoside, 27—naringin, 28—hesperidin, 36—trans-cinnamic acid, 39—apigenin, 40—kaempferol, 1, 4, 6–8, 10–13, 15, 17, 18, 20–22, 26, 29–35, 37, 38, 41–44—unidentified peaks.

**Figure 2 antioxidants-11-02028-f002:**
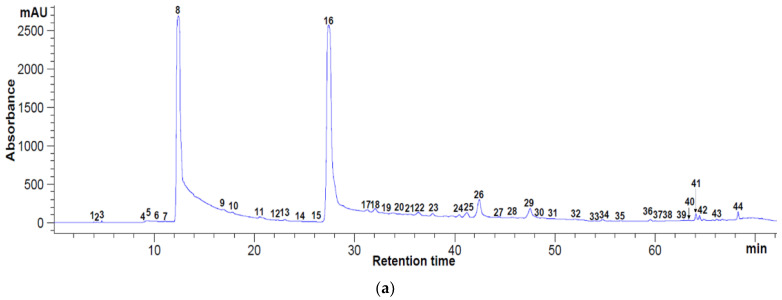
HPLC analysis for bounded polyphenolic compounds in freeze-dried apple pomace extract at 280 nm (**a**) and 320 nm (**b**). Peaks’ identification: (**a**): 5—gallic acid, 21—caffeic acid, 30—quercetin 3—glucoside, 36—hesperidin, 40—myricetin, 1–4, 6–20, 22–29, 31–35, 37–39, 41–44—unidentified peaks; (**b**): 4—caffeic acid, 9—quercetin 3—D-galactoside, 12—hesperidin, 19—luteolin, 20—trans-cinnamic acid, 22—kaempferol; 23—isorhamnetin, 1–3, 5–8, 10, 11, 13–18, 21, 24–28—unidentified peaks.

**Table 1 antioxidants-11-02028-t001:** The free and bounded phenolic compounds in freeze-dried apple pomace.

Bioactive Compound	Concentration (mg/100 g DM Freeze-Dried Pomace)	
	Free Fraction	Bounded Fraction
**Phenolic acids**		
Gallic acid	86.42 ± 2.26 ^a^	8.30 ± 0.26 ^b^
Caffeic acid	3.62 ± 0.80 ^b^	4978.00 ± 900.00 ^a^
Ellagic acid	2.90 ± 0.02 ^b^	50.24 ± 0.34 ^a^
Protocatechuic acid	2.88 ± 0.02 ^b^	2144.20 ± 37.60 ^a^
**Flavan-3-ols**		
Theaflavin	71.73 ± 0.72 ^a^	10.23 ± 0.21 ^b^
Catechin	9.98 ± 0.00 ^a^	N.D.
Cafestol	12.96 ± 0.22 ^a^	4.17 ± 0.81 ^b^
Procyanidin A1	5.27 ± 0.00 ^b^	5.77 ± 0.02 ^a^
Procyanidin B1	4.28 ± 0.01 ^a^	1.20 ± 0.17 ^b^
(−)-Epigallocatechin	2.46 ± 0.57 ^a^	N.D.
**Xanthine**		
Caffeine	4.89 ± 0.07 ^a^	1.93 ± 0.03 ^b^
**Flavonols**		
Myricetin	4.84 ± 0.01 ^a^	N.D.
Quercetin 3-glucoside	0.40 ± 0.00 ^a^	N.D.
Quercetin 3-D-galactoside	N.D.	22.18 ± 0.10 ^a^
Quercetin 3-β-D-glucoside	N.D.	236.60 ± 3.12 ^a^
Isorhamnetin	N.D.	17.06 ± 0.83 ^a^
**Flavanones**		
Hesperidin	42.83 ± 0.27 ^a^	N.D.
Naringin	2.64 ± 0.24 ^a^	N.D.
Apigenin	N.D.	3.18 ± 0.12 ^a^
Kaempferol	N.D.	27.80 ± 1.36 ^a^
Luteolin	N.D.	0.86 ± 0.05 ^a^

Values in the same row that do not share a letter (^a, b^) are statistically different (*p* < 0.05) according to the Tukey test (95% confidence level). N.D.—not determined.

**Table 2 antioxidants-11-02028-t002:** Potential prebiotic effect of freeze-dried apple pomace supplementation of MRS broth for *Loigolactobacillus bifermentans* MIUG BL 16.

	Prebiotic Effect		
Storage (Days)	Supplementation Ratio, % (*w*/*v*)		
	0.5	1	2
0	0	0	0
7	1.50 ± 0.02 ^a^	1.53 ± 0.07 ^a^	1.54 ± 0.08 ^a^
14	3.22 ± 0.12 ^a^	3.63 ± 0.19 ^b^	3.87 ± 0.22 ^ab^
21	3.99 ± 0.56 ^a^	4.58 ± 0.41 ^b^	4.74 ± 0.21 ^c^

Data are showed as mean ± SD (n = 3). Values in the same row that do not share a letter (^a, b, c^) are statistically different (*p* < 0.05) according to the Tukey test (95% confidence level).

**Table 3 antioxidants-11-02028-t003:** Global phytochemical content, antioxidant activity and viable cell counts of the ingredients.

Compound (/g DM)	Variant 1	Variant 2	Variant 3
Total polyphenolic content (mg GAE)	5.32 ± 0.06 ^c^	5.64 ± 0.09 ^b^	6.38 ± 0.14 ^a^
Total flavonoids content (mg CE)	4.02 ± 0.09 ^b^	5.21 ± 0.35 ^a^	5.59 ± 0.22 ^a^
Antioxidant activity (mMol Trolox)	22.02 ± 2.11 ^b^	23.01 ± 1.37 ^b^	42.25 ± 4.58 ^a^
Viable cells (CFU)	3.0 × 10^5 c^	2.05 × 10^6 a^	1.10 × 10^6 b^

Variant 1—freeze-dried apple pomace dissolved in sterile distillated water (20%) and 1% probiotic culture; Variant 2—freeze-dried apple pomace dissolved in sterile distillated water (20%) with 2% inulin and 1% probiotic culture; Variant 3—freeze-dried powder containing concentrated extract in biopolymeric solution of soy protein isolate (2%), inulin (1%) and 1% probiotic culture. Values in the same row that do not share a letter (^a, b, c^) are statistically different (*p* < 0.05) according to the Tukey test (95% confidence level).

**Table 4 antioxidants-11-02028-t004:** Global phytochemical content, antioxidant activity and viable cell counts of value-added fermented soya milk.

Compound (/100 g DM)	C	S1	S2	S3
Total polyphenolic content (mg GAE)	49.96 ± 1.92 ^b^	53.01 ± 1.63 ^ab^	51.17 ± 0.33 ^ab^	56.49 ± 2.00 ^a^
Total flavonoids content (mg CE)	42.84 ± 2.44 ^b^	47.05 ± 2.44 ^b^	46.54 ± 1.19 ^b^	55.31 ± 1.38 ^a^
Antioxidant activity (mMol Trolox)	22.08 ± 3.23 ^c^	30.06 ± 3.71 ^bc^	37.95 ± 1.31 ^ab^	40.23 ± 1.27 ^a^
Viable cells (CFU)/g DM	5.52 × 10^6 d^	1.58 × 10^7 c^	4.88 × 10^7 a^	3.82 × 10^7 b^

C—fermented soya milk without powder addition; S1—fermented soya milk with addition of 2% from powder V1; S2—fermented soya milk with addition of 2% from powder V2; S3—fermented soya milk with addition of 2% from powder V3. Values in the same row that do not share a letter (^a, b, c, d^) are statistically different (*p* < 0.05) according to the Tukey test (95% confidence level).

## Data Availability

Not applicable.

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
