# Peer review of "Upgrading the Functional Potential of Apple Pomace in Value-Added Ingredients with Probiotics"

_antioxidants, 2022, doi:10.3390/antiox11102028_

Round 1
Reviewer 1 Report
I attach documents with corrections.

Author Response
GENERAL COMMENTS OF THE AUTHORS: The authors thank reviewers for the comments aimed to improve our work. The manuscript was revised according to reviewers recommendations, by adding additional information if needed. The corrections are given in different colors in the revised form of the manuscript, depending on the reviewer.
REVIEWER 1:
Authors answer: The authors thank reviewer for the comments. The manuscript was revised according to reviewer recommendations, by adding additional information if needed, highlighted in red. The corrections are given in the revised manuscript.
Reviewer 2 Report
The manuscript titled “Upgrading the functional potential of apple pomace in value-2 added ingredients with probiotics” is interesting and apport new strategies in the development of new foods. However, for clearer authors must include information in the introduction section, and rewrite some section of material and methods, and discusses in deep the results. For example, link the content of bound phenolic compounds found in apple samples with the possible prebiotic effects on the gut microbiome (I suggest the reading of “Caffeic acid prevents non-alcoholic fatty liver disease induced by a high-fat diet through gut microbiota modulation in mice” https://doi.org/10.1016/j.foodres.2021.110240), liked the esterification degree of pectin with its prebiotic activity (I suggest the reading of “Potential of Pectins to Beneficially Modulate the Gut Microbiota Depends on Their Structural Properties” https://doi.org/10.3389/fmicb.2019.00223), or the already use of apple in the industry and the different by-products (I suggest the reading of “Improving oxidative stability of foods with apple-derived polyphenols” 10.1111/1541-4337.12869.
Keywords
I suggest using different keywords. Since selected keywords are already included in the abstract.
1. Introduction
The introduction section explains clearly the problem to study. However, I suggest to included information about the studied probiotic.
2. Material and methods.
In the fruit processing section is not clear if the peel of the apple is included or not. Moreover, in the food industry before making juice pulp is treated with lemon and honey. Even, why use honey?
It is very confusing about the method to extract polyphenols, especially the free fraction due to two different methods has been included in the manuscript. First in the section 2.3. explain the extraction methods using ethanol and ultrasonic, and then in the section 2.5 inside to chromatographic analysis of polyphenols is included that the free fraction was extracted with methanol-agitation.
I proposed, if I understand well the following structure for be clearer and avoid confusions.
If the first method (ethanol-ultrasonic) is only used by spectrophotometric determinations specified this in the methods.
Then, I suggest to include a new section explain the extraction of free and bound polyphenols for polyphenols profile.
Finally, why to use two different methods? Why not determined TPC and TFC to free and bound fractions? Or only to the free fraction?
Section 2.4. Including references to the methods. Moreover, I encourage to use of the “standardized” abbreviation for total phenol content (TPC) and total flavonoid content (TFC).
Section 2.7 Including references to the method
Section 2.8. Including the name of the reactivated bacterial strain.
Section 2.11 I suggest to use of carrier terminology instead to powders, since it is often use by researches when use a fiber or ingredient to “store” a bacterium.
Line 54-55. Rephrasing for not repeat words like source or rich.
Line 67. Change “reduces” by an adjective
Line 70-72. In this paragraph it not be explained the relation between probiotic viability with prebiotics.
Line 107. Check if correct the word “de” or is a mistake
Line 264. Change fro, to from
Figure 1. I encourage you to move letters inside to the chromatography instead above.
Figure 1. Have all detected peaks their spectrum?? Many times, the peaks do not have a spectrum and therefore it's difficult to know if the peak correspond to polyphenol compounds, or other compounds like sugars, organic acids or even aromatic amino acids. Therefore, I encourage the authors to put only numbers in the peaks with the available spectrum, since in my experience, I doubt that all the peaks in the chromatographs have available spectrum and therefore be possible to elucidate if are polyphenols, or other compounds.
Table 1I encourage the authors to include only the maximum contraction detected by the compounds and not separate them by wavelength. It is normal that some compounds absorb energy to different wavelengths. Due to that, it's frequently to use various wavelengths, 280, 320, 360, and 520 to have a complete range where (poly)phenols compounds absorb energy. Even I encourage authors to include 360 nm wavelength in their chromatographic method, to detect many flavonoids that only have the absorbance to this wavelength.
Figure 2. What about peak 8 and 16?
Nowadays for consideration to an ingredient or compound a prebiotic it must be carried out many in vivo determinations have been carried out: “It was implicit that trials to demonstrate prebiotic effects should be performed in the target host. In vitro assessments designed to identify pathways or mechanisms would not confirm prebiotic status in the absence of studies providing evidence of health effects in the host. since intestinal is a complex ecosystem and prebiotics is well defined to was kind of bacterium favorable” Gibson, 2017 Expert consensus document: The International Scientific Association for Probiotics and Prebiotics (ISAPP) consensus statement on the definition and scope of prebiotics. https://doi.org/10.1038/nrgastro.2017.75 Therefore I suggest using the term “potential prebiotic” or even carbohydrate source to Loigolactobacillus bifermentans grow.
Why in the Table 3 is not included control sample?
Line 439. Currently, as you explain polyphenols star to be considered prebiotics since they can modulate the microbiota intestine. In your case, you only tested the growth of the bacterium, therefore, the growth was also stimulated by pectin, sugar or even polyphenols.
Line 444. What is a bioingredient?
Line 447 and line 474. I suggest replacing the word “profile” with content or a similar word, since "profile" is more correct to use when a global characterization of the individual compounds, or when all spectrum of phytochemicals (carotenoids, alkaloids, polyphenols, etc) was determined.
Line 504. Specific that is in the bound fraction
Line 507 cursive the name of bacteria
Line 512. Satisfactory is not an objective word. Why the content of phytochemical is satisfactory, satisfactory for what? Improve health, improve taste, promote lactobacillus grown? Be more specific.
Author Response
GENERAL COMMENTS: The manuscript titled “Upgrading the functional potential of apple pomace in value added ingredients with probiotics” is interesting and apport new strategies in the development of new foods. However, for clearer authors must include information in the introduction section, and rewrite some section of material and methods, and discusses in deep the results. For example, link the content of bound phenolic compounds found in apple samples with the possible prebiotic effects on the gut microbiome (I suggest the reading of “Caffeic acid prevents non-alcoholic fatty liver disease induced by a high-fat diet through gut microbiota modulation in mice” https://doi.org/10.1016/j.foodres.2021.110240), liked the esterification degree of pectin with its prebiotic activity (I suggest the reading of “Potential of Pectins to Beneficially Modulate the Gut Microbiota Depends on Their Structural Properties” https://doi.org/10.3389/fmicb.2019.00223), or the already use of apple in the industry and the different by-products (I suggest the reading of “Improving oxidative stability of foods with apple-derived polyphenols” 10.1111/1541-4337.12869
Authors answer: The authors thank reviewer for the comments. The manuscript was revised according to reviewer recommandations, by adding additional information if needed, in blue color.
COMMENT 1: Keywords - I suggest using different keywords. Since selected keywords are already included in the abstract.
Authors answer: Thank you for your suggestion. Different keywords were added, such as:
Line 29-30: Keywords: apple pomace; probiotics; prebiotics; pectin; value-added; polyphenols, circular economy.
Authors answer: Thank you for your suggestion. Additional information was added, as follow:
Lines 461-469: The prebiotic effect of polysaccharides is related to probiotics ability to degrade pectins, and/or utilize them and other metabolites, by cross-feeding interactions [42]. Larsen et al. [42] suggested that one of the most important factors in prebiotic effect of pectin is the degree of esterification of polygalacturonic acid, with an increase in microbiota composition for the low methoxyl pectins. Therefore, the significant prebiotic effect of freeze-dried apple pomace could be explained by the low methoxyl content for the pectin determined in this study (5.58±0.88%). However, these authors highlighted the possibility of differential stimulation of bacterial populations using pectins with different sugar content [42].
References:
- Larsen, N.; Bussolo de Souza, C.; Krych, L.; Barbosa Cahú, T.; Wiese, M.; Kot, W.; Hansen, K.M.; Blennow, A.; Venema, K.; Jespersen, L. Potential of Pectins to Beneficially Modulate the Gut Microbiota Depends on Their Structural Properties. Front. Microbiol. 2019, 10, 223.
COMMENT 2: 1. Introduction. The introduction section explains clearly the problem to study. However, I suggest to included information about the studied probiotic.
Authors answer: Thank you for your suggestion. Information about the LAB was introduced at lines:
Lines 91-100: The vital role of the lactic acid bacteria (LAB) in human health [21] is well studied, with a significant dimension of industrial applications, both in the health and agri-food industries to enhance food quality and human well-being [22]. Recently, the LAB taxonomy included 261 species, divided into 26 genera based on their whole genome sequences [23]. Significant scientific data support the benefits of lactic acid bacteria as probiotics, modulating the gut microbiota due to they ability to compete with pathogens, immunomodulatory, anti-obesity, anti-diabetic, and anti-cancer activities [24]. Loigolactobacillus bifermentans (Lo. bifermentans) is a facultatively heterofermentative lactic acid bacterium, generally isolated from cheeses, with a potential for fer-mentation of lactic acid into acetic acid, ethanol, traces of propionic acid, carbon di-oxide, and hydrogen [25].
References:
- Heeney, D.D.; Gareau, M.G.; Marco, M.L. Intestinal Lactobacillus in health and disease, a driver or just along for the ride? Curr. Opin. Biotechnol. 2018, 49, 140–147.
- Mota-Gutierrez, J.; Cocolin, L. Current trends and applications of plant origin lactobacilli in the promotion of sustainable food systems. Trends Food Sci. Technol. 2021, 114, 198–211.
- Zheng, J.; Wittouck, S.; Salvetti, E.; Franz, C.M.A.P.; Harris, H.M.B.; Mattarelli, P.; et al. A taxonomic note on the genus Lactobacillus: Description of 23 novel genera, emended description of the genus Lactobacillus Beijerinck 1901, and union of Lactobacillaceae and Leuconostocaceae. Int. J. Syst. Evol. Microbiol. 2020, 70(4), 2782–2858.
- Das, T.K.; Pradhan, S.; Chakrabarti, S.; Mondal, K.C.; Ghosh, K. Current status of probiotic and related health benefits. Appl. Food Res. 2022, 2, 100185.
- McMahon, D.J.; Bowen, I.B.; Green, I.; Domek, M.; Oberg, C.J. Growth and survival characteristics of Paucilactobacillus wasatchensis WDCO4. J. Dairy Sci. 2020, 103, 8771–8781.
COMMENT 3. 2. Material and methods. In the fruit processing section is not clear if the peel of the apple is included or not. Moreover, in the food industry before making juice pulp is treated with lemon and honey. Even, why use honey?
Authors comment: The reviewer is right and we want to thank for the observation. However, our study may be considered as a proposal customized variant for industrial applications. The lemon juice and honey may be replaced by ascorbic acid and sugar solution. However, in order to bring more additional information, some information was added in the revised manuscript, as follow:
Lines 141-144: The composition of the apple’s immersion solution took into account the use of natural sources of ascorbic acid and sugars, such as lemon juice and honey. Ascorbic acid is a common antioxidant that can rapidly reduce quinones and inhibit enzymatic brown-ing [26].
References:
- Zhou, L.; Liao, T.; Liu, W.; Zou, L.Q.; Liu, C.M.; Terefe, N.S. Inhibitory effects of organic acids on polyphenol oxidase: From model systems to food systems. Crit. Rev. Food Sci. Nutr, 2020, 60(21), 3594–3621.
COMMENT 4: It is very confusing about the method to extract polyphenols, especially the free fraction due to two different methods has been included in the manuscript. First in the section 2.3. explain the extraction methods using ethanol and ultrasonic, and then in the section 2.5 inside to chromatographic analysis of polyphenols is included that the free fraction was extracted with methanol-agitation. I proposed, if I understand well the following structure for be clearer and avoid confusions. If the first method (ethanol-ultrasonic) is only used by spectrophotometric determinations specified this in the methods. Then, I suggest to include a new section explain the extraction of free and bound polyphenols for polyphenols profile.
Authors answer: Thank you for your suggestion. Additional information was added, as follow:
Lines 151-152: In order to have an overall view of the global polyphenols content, a conventional solid-liquid solvent ultrasound assisted method was applied.
Lines 158-160: The obtained extract was used for the spectrophotometric analysis of total polyphenolic (TPC) and flavonoids contents (TFC).
Lines 180-189: 2.5. Extraction of free and bounded polyphenols from freeze-dried apple pomace
The free and bounded bioactive compounds’ extraction and separation was performed as described by Xiong et al. [28], slightly modified as Zhang et al. [29]. Briefly, the freeze-dried apple pomace’s free biactives were extracted with methanol (80% v/v) at 25°C and 150 rpm for 2 h (Lab Companion SI-300, GMI, USA), the supernatant being afterwards concentrated. For the bounded bioactive compounds from the analyzed sample, the pellet resulted after free bioactives’ extraction was mixed with 2M HCl and hydrolized at 100°C for 1 h. Thereafter, the bounded bioactive compounds were extracted with ethyl acetate and then concentrated. The obtained fractions were used for chromatographic analysis of the polyphenols.
Lines 190: 2.6. Cromatographic analysis of polyphenols from freeze-dried apple pomace
The separation and identification of the bioactive compounds from freeze-dried apple pomace was performed by an Agilent 1200 HPLC system equipped with autosampler, degasser, quaternary pump system, multi-wavelength detector (MWD) and column thermostat (Agilent Technologies, Santa Clara, CA, USA). A Synergi Max-RP-80 Å column (250 × 4.6 mm, 4 µm particle size, Phenomenex, Torrance, CA, USA) was used for the free and bounded polyphenols analysis. The concentrated samples, respectively free and bouded fraction, were dissolved in methanol and subjected to the HPLC separation at 30°C using 10 µL injection volume, with solvent A (1% v/v acetic acid in ultrapure water) and solvent B (acetonitrile) using a flow rate of 0.65 mL/min. The method runtime was 80 min, the compounds of interest were detected at 280 nm and 320 nm, using a 10 nm bandwidth for each detection wavelength to increase compounds’ selective separation. The identification of the bioactive compounds from apple pomace was made by comparing the retention times of the peaks with those obtained for standard solutions of bioactives. The identified compounds were quantified by external calibration curves using the peak area. Data acquisition was performed by Chemstation software, version B.04.03 (Agilent Technologies, Santa Clara, CA, USA). Results were expressed in mg/100 g DW freeze-dried pomace.
COMMENT 5: Finally, why to use two different methods? Why not determined TPC and TFC to free and bound fractions? Or only to the free fraction?
Authors comment: The free and bounded fractions were employed only in the HPLC analysis, considering that the chromatographic separation allows an improved separation and identification of the compounds previously determined by spectrophotometric methods. It should be mentioned that the bioactives were previously determined by HPLC methods from the ethanolic extracts used for the spectrophotometric assays, the obtained results being unsatisfactory (data not reported).
COMMENT 6: Section 2.4. Including references to the methods. Moreover, I encourage to use of the “standardized” abbreviation for total phenol content (TPC) and total flavonoid content (TFC).
Authors answer: The corrections were made in the revised form of the manuscript. The references were included as follow:
References:
- Turturică, M.; Stănciuc, N.; Bahrim, G.; Râpeanu, G. Effect of thermal treatment on phenolic compounds from plum (prunus domestica) extracts – A kinetic study. J. Food Eng. 2016, 171, 200-207.
COMMENT 7: Section 2.7 Including references to the method.
Authors answer: The references were included as follow:
References:
- Turturică, M.; Stănciuc, N.; Bahrim, G.; Râpeanu, G. Effect of thermal treatment on phenolic compounds from plum (prunus domestica) extracts – A kinetic study. J. Food Eng. 2016, 171, 200-207.
COMMENT 8: Section 2.8. Including the name of the reactivated bacterial strain.
Authors answer: The corrections were made in the revised form of the manuscript
Line 249: 2.9. Lo. bifermentans MIUG 16 reactivation and inoculum preparation.
Line 253: Lo. bifermentans MIUG 16
COMMENT 9: Section 2.11 I suggest to use of carrier terminology instead to powders, since it is often use by researches when use a fiber or ingredient to “store” a bacterium.
Authors answer: The corrections were made in the revised form of the manuscript.
COMMENT 10: Line 54-55. Rephrasing for not repeat words like source or rich.
Authors answer: The corrections were made in the revised form of the manuscript.
Lines 55-56: The apple pomace is rich in minerals, dietary fiber and polyphenols, and pectin
COMMENT 11: Line 67. Change “reduces” by an adjective
Authors answer: The corrections were made in the revised form of the manuscript, as follow:
Lines 76-77: and reducing diarrhea incidence and blood cholesterol
COMMENT 12: Line 70-72. In this paragraph it not be explained the relation between probiotic viability with prebiotics.
Authors answer: Some additional information were added in the revised form of the manuscript, as follow:
Lines 82-86: The prebiotic mechanism is explained by their slow fermentation from complex structures, thus providing fermentable carbohydrates for bacteria in the distal colon, allowing to regulate the dysbiosis of gut microbiota, whereas producing metabolic butyrate to modulate gut barrier function and anti-inflammatory effect [18].
References:
- Ferreira-Lazarte, A.; Moreno, F.J.; Cueva, C.; Gil-Sanchez, I.; Villamiel, M. Behaviour of citrus pectin during its gastrointestinal digestion and fermentation in a dynamic simulator (simgi(R)). Carbohydr. Polym. 2019, 207, 382–390.
COMMENT 13: Line 107. Check if correct the word “de” or is a mistake
Authors answer: The name of the cultivation medium is correct.
COMMENT 14: Line 264. Change fro, to from
Authors answer: The corrections were made in the revised form of the manuscript.
COMMENT 15: Figure 1. I encourage you to move letters inside to the chromatography instead above.
Authors answer: The letters above the panels from Figure 1 and 2 were included inside them, as you suggested.
COMMENT 16: Figure 1. Have all detected peaks their spectrum?? Many times, the peaks do not have a spectrum and therefore it's difficult to know if the peak correspond to polyphenol compounds, or other compounds like sugars, organic acids or even aromatic amino acids. Therefore, I encourage the authors to put only numbers in the peaks with the available spectrum, since in my experience, I doubt that all the peaks in the chromatographs have available spectrum and therefore be possible to elucidate if are polyphenols, or other compounds.
Authors answer: The presence of other compounds in the analyzed sample by HPLC is possible, due to the complexity of the food matrix involved, respectively apple pomace. Therefore, the experimental data presented in the manuscript refer only to polyphenols, because:
- The organic acids are often determined at 210 nm;
- The sugars must be subjected to a derivatization in order to be detected by UV detectors, without additional Refractive Index Detectors (RID). Even though, detection must be carried out at 245 nm;
- For the separation of amino acids a Fluorescence Detector (FLD) and derivatization procedures must be employed in order to properly identify these compounds, using excitation and emission wavelengths ranging between 220-250 nm, respectively 450-470 nm.
The separation method of the reported bioactives from this work was a correct choice, considering the previously mentioned aspects, as well as paying attention to the solvents used, the chromatographic column, and detection wavelengths. Moreover, the bioactives’ selective separation was improved, for the selected wavelengths, by a restricted bandwidth allocation of 10 nm.
COMMENT 17: Table 1. I encourage the authors to include only the maximum contraction detected by the compounds and not separate them by wavelength. It is normal that some compounds absorb energy to different wavelengths. Due to that, it's frequently to use various wavelengths, 280, 320, 360, and 520 to have a complete range where (poly)phenols compounds absorb energy. Even I encourage authors to include 360 nm wavelength in their chromatographic method, to detect many flavonoids that only have the absorbance to this wavelength.
Authors answer: As you suggested, only the maximum concentrations detected were presented in Table 1. Also, the authors decided to combine Table 1 and Table 2. The most relevant experimental data were reported in this manuscript. Thank you very much for this supportive comment regarding the wavelengths to be included in the chromatographic method, we will consider this for our future works.
COMMENT 18: Figure 2. What about peak 8 and 16?
Authors answer: Those peaks were between the unidentified compounds, as it was mentioned in Figure 2 legend: 1 – 3, 5 – 8, 10, 11, 13 – 18, 21, 24 – 24 – unidentified peaks.
COMMENT 19: Nowadays for consideration to an ingredient or compound a prebiotic it must be carried out many in vivo determinations have been carried out: “It was implicit that trials to demonstrate prebiotic effects should be performed in the target host. In vitro assessments designed to identify pathways or mechanisms would not confirm prebiotic status in the absence of studies providing evidence of health effects in the host. since intestinal is a complex ecosystem and prebiotics is well defined to was kind of bacterium favorable” Gibson, 2017 Expert consensus document: The International Scientific Association for Probiotics and Prebiotics (ISAPP) consensus statement on the definition and scope of prebiotics. https://doi.org/10.1038/nrgastro.2017.75 Therefore I suggest using the term “potential prebiotic” or even carbohydrate source to Loigolactobacillus bifermentans grow.
Authors answer: The corrections were made all over the manuscript, by replacing the word prebiotic with potential prebiotic.
COMMENT 20: Why in the Table 3 is not included control sample?
Authors answer: The data included in Table 2 in the revised form of the manuscript are calculated based on equation 5, considering the control samples.
COMMENT 21: Line 439. Currently, as you explain polyphenols star to be considered prebiotics since they can modulate the microbiota intestine. In your case, you only tested the growth of the bacterium, therefore, the growth was also stimulated by pectin, sugar or even polyphenols.
Authors answer: Thank you for your suggestion. Corrections were made at lines xxx, as follow: Although the prebiotic effect of apple pomace was clearly evidenced in this study, the potential role of polyphenols, sugars and pectin as modulators of gut microbiota should be further tested.
COMMENT 22: Line 444. What is a bioingredient?
Authors answer: Thank you for your comment. The word bioingredient was replaced by ingredient all over the manuscript. In the first hypothesis, we considered the powders containing probiotics as bioingredients.
COMMENT 23: Line 447 and line 474. I suggest replacing the word “profile” with content or a similar word, since "profile" is more correct to use when a global characterization of the individual compounds, or when all spectrum of phytochemicals (carotenoids, alkaloids, polyphenols, etc) was determined.
Authors answer: Thank you for your comment. The word profile was replaced by content, when appropriate.
COMMENT 24: Line 504. Specific that is in the bound fraction
Authors answer: Thank you for your comment. Additional information was added, as follow:
Line 546-550: The gallic acid, theaflavin and hesperidin were the major free compounds separated from the methanolic fraction, whereas caffeic acid showed the highest concentration of approximatively 50 mg/g DM, followed by trans-cinnamic acid, and quercetin 3-β-D-glucoside, these compounds being identified from the bounded fraction.
COMMENT 25: Line 507 cursive the name of bacteria
Authors answer: Thank you for your comment. Correction was made at line 554.
COMMENT 26: Line 512. Satisfactory is not an objective word. Why the content of phytochemical is satisfactory, satisfactory for what? Improve health, improve taste, promote lactobacillus grown? Be more specific.
Authors answer: Thank you for your comment. Correction was made as follow:
Lines 558-559: The ingredients were analysed for the phytochemical content and viable cells count of at least 6 log/g.
Reviewer 3 Report
The manuscript of Camelia Cristina Vlad et al. describes the possible valorization of apple pomace as a functional ingredient with probiotic action in foods. The work is well conceived and developed, unfortunately, in some respects, it is at a very preliminary level. This should be highlighted starting with the title. The major concerns, in this sense, become evident in many lines of the text. In the abstract (lines 11-12) the Authors underline the need for sustainable methodologies for the transformation of food industry by-products into bioactive ingredients. Furthermore, in line 495 the Authors affirm that the experimental design proposed by them is sustainable. From this point of view, freeze-drying cannot be considered an eco-sustainable approach and the fact that this is a simplification applicable mainly at the laboratory level should be highlighted. In line 111, the authors also describe that seeds and stems have been removed manually, I presume, even if this should be specified. This operation, useful in the laboratory, cannot be reproduced at the industrial level and also modifies the phenolic composition of the final pomace. All the subsequent operations described (lines 111-115) for the production of apple pomace are also a model on a laboratory scale which, however, can modify the composition of the final product and cannot represent the real industrial by-product. The characterization of the phytochemical profile (lines 273 and following) is carried out with a slightly outdated method (HPLC). As consequence, numerous chromatographic peaks are not identified. Also from this point of view, the preliminary aspect of the work should be highlighted by suggesting to the reader that future studies must be conducted. In lines 344-345, the same Authors highlight how freeze-drying can change the phenolic profile of apple pomace. This limitation of the approach should therefore be highlighted in the text. In addition, in lines 381-383 and 387-388, the Authors declare that the removal of seeds and stems makes their pomace different from the real industrial by-product. At this point, the reader wonders what the significance of the results presented is. In lines 407 and following, the authors describe a potential prebiotic effect of the matrix but, again, a further study on simulated digested samples would be necessary. Finally, in lines 510-511 the Authors speak about a microencapsulated product. If the Authors intend to use terms like this, a minimal physical characterization of the matrix should be included. Minor considerations: Why do the panels of figures 1 and 2 all have different scales? Line 507: "Loigolactobacillus bifermentans" should be written in italics.
Author Response
REVIEWER 3:
COMMENTS: The manuscript of Camelia Cristina Vlad et al. describes the possible valorization of apple pomace as a functional ingredient with probiotic action in foods. The work is well conceived and developed, unfortunately, in some respects, it is at a very preliminary level. This should be highlighted starting with the title. The major concerns, in this sense, become evident in many lines of the text.
Authors answer: The authors thank reviewer for the comments. The manuscript was revised according to reviewer recommandations, by adding additional information if needed, in green color. All the issues raised by the reviewer were addressed point by point.
COMMENT 1: In the abstract (lines 11-12) the Authors underline the need for sustainable methodologies for the transformation of food industry by-products into bioactive ingredients. Furthermore, in line 495 the Authors affirm that the experimental design proposed by them is sustainable. From this point of view, freeze-drying cannot be considered an eco-sustainable approach and the fact that this is a simplification applicable mainly at the laboratory level should be highlighted.
Authors answer: The authors thank reviewer for the proper comment. The abstract was reconsidered, as follow:
Lines 11-13: Emerging customized designs to upgrade the functional potential of freeze-dried apple pomace was used in this study, in order to transform the industrial by-products into ingredients containing probiotics, for a better and healthier food composition.
Lines 539-540: Therefore, in our study, different customized designs to valorize apple pomace into ingredients were studied…
Lines 561-564: The results obtained in this study unveil a readily available method to straightforward processes in order to upgrade the great potential of the fruits by-products by transforming into value-added ingredients for different applications.
COMMENT 3: In line 111, the authors also describe that seeds and stems have been removed manually, I presume, even if this should be specified. This operation, useful in the laboratory, cannot be reproduced at the industrial level and also modifies the phenolic composition of the final pomace. All the subsequent operations described (lines 111-115) for the production of apple pomace are also a model on a laboratory scale which, however, can modify the composition of the final product and cannot represent the real industrial by-product.
Authors answer: The authors thank reviewer for the proper comment. An additional information was added, as follow:
Lines 141-144: The composition of the apple’s immersion solution took into account the use of natural sources of ascorbic acid and sugars, such as lemon juice and honey. Ascorbic acid is a common antioxidant that can rapidly reduce quinones and inhibit enzymatic browning [26].
References:
26.Zhou, L.; Liao, T.; Liu, W.; Zou, L.Q.; Liu, C.M.; Terefe, N.S. Inhibitory effects of organic acids on polyphenol oxidase: From model systems to food systems. Crit. Rev. Food Sci. Nutr, 2020, 60(21), 3594–3621.
COMMENT 4: The characterization of the phytochemical profile (lines 273 and following) is carried out with a slightly outdated method (HPLC). As consequence, numerous chromatographic peaks are not identified. Also from this point of view, the preliminary aspect of the work should be highlighted by suggesting to the reader that future studies must be conducted.
Authors answer: The HPLC analysis allowed a preliminary characterization of bioactives from apple pomace, which is an important starting point to emphasize the benefits and functionality of this by-product, respectively of the novel value-added designed functional foods reported in this work. Taking into consideration your comment, some information regarding the preliminary chromatographic analysis and further studies to be carried out in the future was stated in the manuscript, as follows:
Line 543: “The preliminary chromatographic analysis evidenced the presence of both free and bonded polyphenolic compounds (…)”.
Line 550-552: “Therefore, an in-depth bioactives’ characterization along with their interactions and mechanisms of action must be considered to emphasize and understand the functionality of the designed products with apple pomace.”
COMMENT 5: In lines 344-345, the same Authors highlight how freeze-drying can change the phenolic profile of apple pomace. This limitation of the approach should therefore be highlighted in the text.
Authors answer: Thank you for your comment. However, the authors draw attention that the statement “Different studies reported that in fresh apple pomace, phenolic compounds are dominated by chlorogenic acid, caffeic acid, (+)-catechin, (−)-epicatechin, rutin, and quercetin glycosides [24], whereas, after freeze-drying, phlorizin (phloretin-2´-β-D-glucopyranoside) is the most prominent polyphenol [25]” are in fact comparisons with the literature (Schieber, A.; Hilt, P.; Streker, P.; Endre, H.-U.; Rentschler, C.; Carle, R. A new process for the combined recovery of pectin and phenolic compounds from apple pomace. Innov. Food Sci. Emer. Technol. 2003, 1(4), 99–107.)
COMMENT 6: In addition, in lines 381-383 and 387-388, the Authors declare that the removal of seeds and stems makes their pomace different from the real industrial by-product. At this point, the reader wonders what the significance of the results presented is.
Authors answer: Thank you for your comment. Please, be aware that at the industrial scale, the removal of seeds and stems may be applied. Additionally, the statement showed comparisons with the data from the literature to explain in fact the absence of phlorizin.
COMMENT 7: In lines 407 and following, the authors describe a potential prebiotic effect of the matrix but, again, a further study on simulated digested samples would be necessary.
Authors answer: Thank you for your comment. Corrections were made all over the manuscript, by introducing the term potential prebiotic effect.
COMMENT 8: Finally, in lines 510-511 the Authors speak about a microencapsulated product. If the Authors intend to use terms like this, a minimal physical characterization of the matrix should be included.
Authors answer: Thank you for your comment. Further studies, including bulk density, solubility etc., will be included in our further studies.
COMMENT 9: Minor considerations: Why do the panels of figures 1 and 2 all have different scales? Line 507: "Loigolactobacillus bifermentans" should be written in italics.
Authors answer: Thank you for your comment. Corrections were made all over the manuscript. The panels from Figure 1 and Figure 2 have different scales because the Oy axis (Absorbance) is adjusting automatically based on the maximum absorbance of the separated peaks. The correction was made in the revised form.
Reviewer 4 Report
This is an interesting work dealing with upgrading the functional potential of apple pomace in value-2 added ingredients with probiotics. The topic is interesting for the scientific community and "Antioxidants" readers, and in my opinion, the manuscript will be suitable for publication after major revision. Some aspects regarding the identification of polyphenols and how data is shown need to be addressed.
I suppose that section 3 is "Results and discussion" and not only "Results"
Line 167-168: bioactive substances were quantified by external calibration. How did the authors ensure that there was no matrix effect with the apple pomace? This need to be addressed in the manuscript.
Figure 1. In my opinion, only those compounds that are identified need to be labeled in the figure. Please, do not label non-identified compounds. Indicate in the figure that only those identified were indicated. If compounds are not identified you cannot ensure if these compounds are or not phenolics. Same comment for Figure 2. A peak that was not identified, cannot be considered a phenolic compound, you cannot be sure independently of the sample treatment and the chromatographic separation employed.
Also, you need to address how peak purity was assessed.
Table 1. Indicate that these are the detected compounds, and do not include those that were not detected at all. I do not see the point of including non-detected phenolics in the table. Same comment for Table 2.
Minor changes:
Line 77: "Therefore, the aim of our study was to test..." or "Therefore, the aim of our study was testing..."
Line 277-278: "Figure 1 shows the contents of free phenolics analysed in 277 freeze-dried apple pomace, at different wavelength (280 nm and 320 nm)." Please, rewrite this sentence, the figure shows chromatograms with the detected phenolics, but not the contents of these compounds.
Author Response
GENERAL COMMENTS: This is an interesting work dealing with upgrading the functional potential of apple pomace in value-2 added ingredients with probiotics. The topic is interesting for the scientific community and "Antioxidants" readers, and in my opinion, the manuscript will be suitable for publication after major revision. Some aspects regarding the identification of polyphenols and how data is shown need to be addressed.
Authors answer: The authors thank reviewer for the comments. The manuscript was revised according to reviewer recommandations, by adding additional information if needed, in purple color. All the issues raised by the reviewer were addressed point by point.
COMMENT 1: I suppose that section 3 is "Results and discussion" and not only "Results".
Authors answer: The authors thank reviewer for the comments. Correction was made at line 309.
COMMENT 2: Line 167-168: bioactive substances were quantified by external calibration. How did the authors ensure that there was no matrix effect with the apple pomace? This need to be addressed in the manuscript.
Authors answer: Thank you for your comment. The matrix effect was minimized by using a properly chosen HPLC elution method, including chromatographic column, elution solvents, and detection wavelengths. The bioactives’ separation selectivity was ensured by a bandwidth allocation of 10 nm for the detection signals. This aspect was mentioned in the manuscript, as following:
Line 200-201: The method runtime was 80 min, the compounds of interest were detected at 280 nm and 320 nm, using a 10 nm bandwidth for each detection wavelength to increase compounds’ selective separation.
COMMENT 3:
Figure 1. In my opinion, only those compounds that are identified need to be labeled in the figure. Please, do not label non-identified compounds. Indicate in the figure that only those identified were indicated. If compounds are not identified you cannot ensure if these compounds are or not phenolics. Same comment for Figure 2. A peak that was not identified, cannot be considered a phenolic compound, you cannot be sure independently of the sample treatment and the chromatographic separation employed.
Authors answer: Thank you for your comment. The maximum absorbances for phenolic acids and flavonoids are 280 and 320 nm. Consequently, it is important to highlight the abundance of bioactives originated from the apple pomace reported in Figure 1 and Figure 2, even though some compounds were not identified, they were detected at the analyzed wavelengths and can possibly be bioactive compounds’ derivatives.
COMMENT 4: Also, you need to address how peak purity was assessed.
Authors answer: Thank you for your comment. The purity of the separated peaks was not a major concern of this study, taking into account that we do not intend to use the separated compounds further for medical or pharmaceutical purposes. Therefore, HPLC-grade standards were used to assess the partial identification and quantification. Based on this aspect, it is important to report all the detected compounds eluted at 280 and 320 nm, as it is shown in Figure 1 and Figure 2. We will consider your suggestion about the peaks’ purity for the upcoming scientific works.
COMMENT 5: Table 1. Indicate that these are the detected compounds, and do not include those that were not detected at all. I do not see the point of including non-detected phenolics in the table. Same comment for Table 2.
Authors answer: Thank you for your comment. The data from Tables 1 and 2 were re-organized. Please, see the revised form of the manuscript.
COMMENT 5:
Minor changes:
Line 77: "Therefore, the aim of our study was to test..." or "Therefore, the aim of our study was testing..."
Authors answer: The authors thank reviewer for the comments. Correction was made at line 101.
Line 277-278: "Figure 1 shows the contents of free phenolics analysed in 277 freeze-dried apple pomace, at different wavelength (280 nm and 320 nm)." Please, rewrite this sentence, the figure shows chromatograms with the detected phenolics, but not the contents of these compounds.
Authors answer: Thank you for your comment. The phrase was rephrased as you suggested.
Round 2
Reviewer 3 Report
The Authors deeply revised the manuscript according to the Referees' suggestions. In the present form, the manuscript is acceptable for publication.
Reviewer 4 Report
The authors correctly addressed my revision suggestions. The manuscript is acceptable for publication in its current form.